

# Microbiota plasticity in tilapia gut revealed by meta-analysis evaluating the effect of probiotics, prebiotics, and biofloc

Marcel Martinez-Porchas[1,*], Aranza Preciado-Álvarez[1,*], Francisco Vargas-Albores[1], Martina Hilda Gracia-Valenzuela[2], Francesco Cicala[3], Luis Rafael Martinez-Cordova[4], Diana Medina-Félix[5] and Estefania Garibay-Valdez[1]

[1] Biología de Organismos Acuáticos, Centro de Investigación en Alimentación y Desarrollo, Hermosillo, Sonora, Mexico
[2] Tecnológico Nacional de México/Instituto Tecnológico del Valle del Yaqui, Bácum, Sonora, México
[3] Department of Comparative Biomedicine and Food Science, University of Padova, Legnaro, Province of Padua, Italy
[4] Departamento de Investigaciones Científicas y Tecnológicas, Universidad de Sonora, Hermosillo, Sonora, Mexico
[5] Departamento de Ecología, Universidad Estatal de Sonora, Hermosillo, Sonora, Mexico
[*] These authors contributed equally to this work.

Corresponding author
Estefania Garibay-Valdez, estefania.garibay@ciad.mx

## ABSTRACT

Tilapia species are among the most cultivated fish worldwide due to their biological advantages but face several challenges, including environmental impact and disease outbreaks. Feed additives, such as probiotics, prebiotics, and other microorganisms, have emerged as strategies to protect against pathogens and promote immune system activation and other host responses, with consequent reductions in antibiotic use. Because these additives also influence tilapia's gut microbiota and positively affect the tilapia culture, we assume it is a flexible annex organ capable of being subject to significant modifications without affecting the biological performance of the host. Therefore, we evaluated the effect of probiotics and other additives ingested by tilapia on its gut microbiota through a meta-analysis of several bioprojects studying the tilapia gut microbiota exposed to feed additives (probiotic, prebiotic, biofloc). A total of 221 tilapia gut microbiota samples from 14 bioprojects were evaluated. Alpha and beta diversity metrics showed no differentiation patterns in relation to the control group, either comparing additives as a group or individually. Results also revealed a control group with a wide dispersion pattern even when these fish did not receive additives. After concatenating the information, the tilapia gut core microbiota was represented by four enriched phyla including Proteobacteria (31%), Fusobacteria (23%), Actinobacteria (19%), and Firmicutes (16%), and seven minor phyla Planctomycetes (1%), Chlamydiae (1%), Chloroflexi (1%), Cyanobacteria (1%), Spirochaetes (1%), Deinococcus Thermus (1%), and Verrucomicrobia (1%). Finally, results suggest that the tilapia gut microbiota is a dynamic microbial community that can plastically respond to feed additives exposure with the potential to influence its taxonomic profile allowing a considerable optimal range of variation, probably guaranteeing its physiological function under different circumstances.

# INTRODUCTION

Tilapia species (*Oreochromis* spp.), carp, catfish, and salmon, rank as the most important farmed freshwater fish species (*Cai et al., 2019*) due to their high adaptability and lower demand for fishmeal in their diet (*Gjedrem & Baranski, 2009*). Particularly tilapia is perhaps the cultivable fish species with better tolerance for a wide range of environmental conditions, handling, diets, and crossbreeding (*Trujillo & Carranza, 2022*), features that have allowed its culture around the globe and in diverse production systems. Tilapia are omnivorous and can be fed a variety of feeds, including plant-based (*Ferreira et al., 2020*; *Xuan et al., 2022*) and animal-based (*Amer et al., 2022*; *Kim et al., 2019*) diets, making them a relatively low-cost species to farm. In addition, the tilapia industry has improved welfare in developing countries by delivering benefits such as household incomes, food security, and nutritional value through increased high-quality protein consumption (*Prabu et al., 2019*).

Even though tilapia aquaculture has experienced significant growth in the last two decades due to the above benefits and the biological advantages of the *Oreochromis* genera, several challenges can limit its productivity and profitability, including bacterial, viral, and parasitic diseases that can cause significant mortality and economic losses in tilapia farms (*Van Hai, 2015*). Common diseases in tilapia include *Streptococcus*, *Aeromonas*, and *Edwardsiella* infections. In addition, the use of high-quality feeds is essential for the growth and health of tilapia. However, feed management can be challenging in tilapia aquaculture, as underfeeding can result in reduced growth rates and health problems, while overfeeding can lead to water quality problems. High-quality water management in fish ponds is another concern since it is a major factor determining fish production (*Salama et al., 2006*). Besides, inadequate temperature, pH, and oxygen levels can lead to stress, disease, and reduced growth rates. Tilapia farming can have environmental impacts, including the discharge of nutrients and waste into waterways and the potential for spreading diseases to wild fish populations (*Baccarin & Camargo, 2005*). Sustainable tilapia farming practices that minimize these impacts are becoming increasingly important.

To solve the problems generated by pathogens, antimicrobials and antiparasitics have been used as preventive and corrective measures (*Cao et al., 2022*), but they have a consequent negative impact in the medium and long term on the environment. The antibiotics administration in high doses or throughout long periods has a severe affectation on microbial communities in both the fish and the environment, as well as triggering antibiotic resistance which can even worsen pathogen control (*Budiati et al., 2013*; *Fang et al., 2021*); thus, such strategies could be a double-edged sword with immediate benefits with mid- or long-term negative consequences.

On the other hand, using probiotics in aquaculture emerged more than three decades ago as an alternative strategy qualified as an "environment-friendly treatment" (*Gatesoupe,*

*1999*). From that point on, a plethora of scientific research on the use of probiotics ensued, including different species of microorganisms to be used as probiotics, mixtures of species, carryover forms of probiotics to ensure delivery to the gut, and even obtaining and using products such as paraprobiotics, prebiotics and synbiotics (*Goh et al., 2022*; *Vargas-Albores et al., 2021*). Over time, the evidence demonstrated that probiotics could benefit fish, such as protection against pathogens and activation of the immune system from different pathways (*Hoseinifar et al., 2018*; *Nikiforov-Nikishin et al., 2022*). In tilapia aquaculture, probiotics are typically administered as a feed supplement, either as a single strain or a combination of microbial strains. The most used probiotic bacteria in tilapia aquaculture include *Lactobacillus*, *Bacillus*, and *Lactococcus* (*Cano-Lozano et al., 2022*; *Xia et al., 2018*), which have improved growth, feed conversion, and disease resistance. On the other hand, prebiotics in fish aquaculture is typically administered as a dietary supplement, such as fructooligosaccharides (FOS) or inulin (*Panase et al., 2023*; *Wang et al., 2021b*). The fish do not digest these compounds; instead, they stimulate the growth and activity of beneficial bacteria in the gut, promoting benefits for the fish (*Panase et al., 2023*). Administered as feed additives, probiotics and prebiotics can provide disease resistance stimulating the tilapia's immune system, making them more resistant to bacterial and viral infections (*Mugwanya et al., 2022*). Probiotics have improved the survival rate of tilapia infected with common pathogens such as *Streptococcus agalactiae* and *Aeromonas hydrophila* (*Chen, Liu & Hu, 2019*; *Wang et al., 2021b*). Probiotics and prebiotics can also improve tilapia's growth rate and feed efficiency, leading to more extensive and healthier fish (*Mugwanya et al., 2022*; *Xuan et al., 2022*).

Due to their benefits, probiotics and prebiotics have made their way into the aquaculture industry; however, improvements in growth and health seem to be associated with the role of these elements in maintaining a healthy microbiota. The gut of tilapia contains a complex community of microorganisms that play a critical role in digestion, immunity, and overall health. Prebiotics can also help to establish a healthy gut microbiota by promoting the growth of beneficial bacteria and reducing the colonization of harmful bacteria (*Opiyo et al., 2019*; *Tan, Chen & Hu, 2019*; *Wang et al., 2021b*), supporting the growth of beneficial bacteria by providing a food source. In addition, the mass growth of beneficial bacteria has been stimulated in intensive systems based on biofloc technology, which are characterized by requiring elevated carbon:nitrogen ratios and intense aeration but with insignificant water exchange, reducing the antibiotic use due to the competence generated by the high concentration of aerobic bacteria (*Robles-Porchas et al., 2020*). The sum of all these benefits coincides with a reduction of environmental impact. One of the most important outcomes is the reduction of the reliance on antibiotics, which can lead to the development of antibiotic-resistant bacteria and contribute to the spread of antibiotic residues in the environment (*Mawardi et al., 2023*; *Mugwanya et al., 2022*).

In recent years, high throughput sequencing has revealed in better resolution how probiotics and other microorganisms can influence the gut microbiota of tilapia (*Haygood & Jha, 2018*; *Standen et al., 2015*; *Yu et al., 2019*). However, it is unclear to what extent these microorganisms used for the benefit of fish manage to change the intestinal microbiota, nor how these impact the core microbiota usually detected in tilapia. Several studies have

provided relevant information on the effect of probiotics and prebiotics by observing changes in the composition of the tilapia gut microbiota; therefore, a meta-analysis concatenating the available information from these projects would provide a panoramic view but also more precise, revealing patterns on the effect of probiotics and prebiotics on tilapia. Herein, meta-analyses have been used to evaluate the gut microbiota of terrestrial animals, define the core microbiota, establish microbial biomarkers, and evaluate the effect of dietary components on the gut microbiota (*Holman et al., 2017*; *Holman & Gzyl, 2019*; *Mancabelli et al., 2017*). Here, we aimed to perform a meta-analysis of the tilapia gut microbiota exposed to probiotics, prebiotics, and biofloc treatments to (1) evaluate the effect of such treatments on the gut microbiota of tilapia and (2), define the species' core microbiota and potential bacterial biomarkers.

## MATERIALS & METHODS

### Datasets and preprocessing of Tilapia gut microbiota

A systematic search for published studies was performed on the Web of Science platform using the keyword (Tilapia AND gut AND (microbiome OR microbiota)), as described in the workflow (Fig. 1). As an outcome, 3,584 potentially useful references were recovered (Fig. S1) and organized in an EndNote (https://endnote.com/) database. This database was again filtered using: "(*Tilapia* OR *Oreochromis*) AND (*Microbiome* OR *Microbiota* OR *Metagenome*) AND (*Probiotic* OR *Prebiotic* OR *Biofloc* OR *Additives*)", resulting in 60 papers considered for deeper search (Table S1). The most relevant papers were thoroughly reviewed based and only considered those that: (a) used high throughput sequencing V3, V4, or both hyper-variable regions of the 16S ribosomal RNA (16S rRNA) gene for microbiota taxonomic description; (b) studied the modulation of tilapia gut microbiome by feed additives (probiotic, prebiotic, biofloc); and (c) the sequences are available as NGS metagenomic data (SRA or Bioproject number) and corresponding subject meta-data (up to November 2022). The full-text assessment and screening process was performed by two authors (APA, EGV), and the referee was MMP.

In addition, the SRA database from NCBI was also explored using the term "tilapia gut microbiome" to find available bioprojects without assigned published papers. Only bioprojects studying the effect of feed additives on the tilapia gut microbiome were considered. Thus, using both strategies (references and SRA database), 14 bioprojects with clear relevance, available metadata, and registered sequencing data were selected (Table S2). Finally, studies that fulfilled the meta-analysis criteria were evaluated for sample type (probiotic, prebiotic, biofloc, and control) and addressed other relevant variables (Age, Additive component, Environment, Gut section, and Geographic location), as described in Table S3.

### Data retrieval and quality control of sequenced reads

Raw sequence files were downloaded from the Sequence Read Archive at NCBI using the SRA Toolkit. A total input of 13,123,343 demultiplexed raw data sequences corresponding to the 16S rRNA hyper-variable region were imported and processed with the Quantitative Insights Into Microbial Ecology 2 (QIIME2), version 2022.2 (*Bolyen et al., 2019*). As data

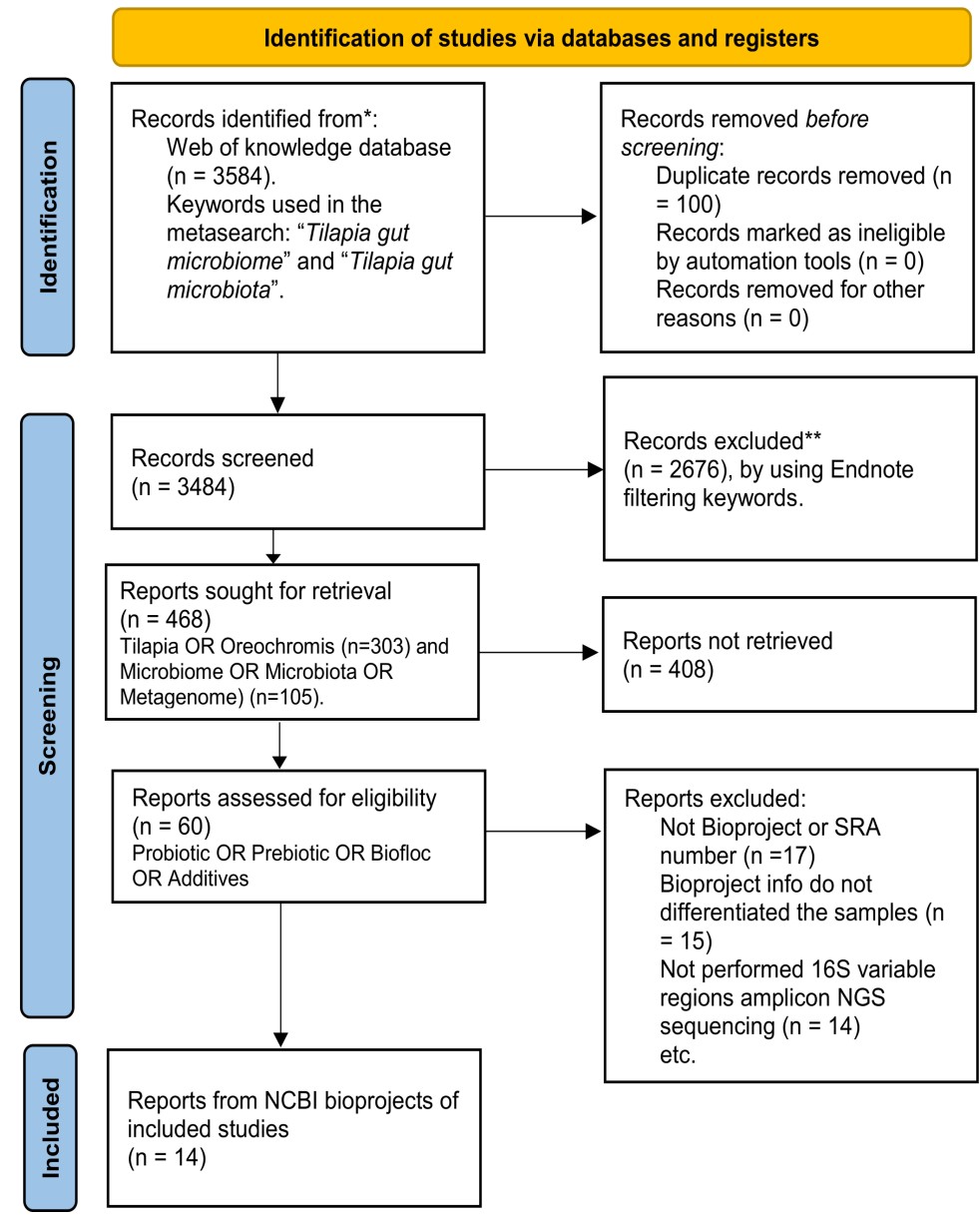

**Figure 1** **PRISMA flow chart for studies and bioprojects inclusion performed during the metasearch.**
Metasearch was performed in the Web of Knowledge platform and 3,584 studies were retrieved, which
were screened. Studies were filtered by using endnote automated tools and keywords. Then, 60 studies
were considered for the deeper search of bioprojects. After screening, we include 14 studies from 14 NCBI
bioprojects. *Consider, if feasible to do so, reporting the number of records identified from each database
or register searched (rather than the total number across all database/registers). **If automation tools were
used, indicate how many records were excluded by a human and how many were excluded by automation
tools.

were mined from different sources, sequences were imported into QIIME2 using the
manifest file (*Estaki et al., 2020*). Raw sequences were preprocessed using an initial quality
filtering process based on quality scores and setting the quality-filter plugin (*Bokulich et al.,*

*2013*). Then, the deblur plugin was used to apply the denoise-16S method to the sequences (*Amir et al., 2017*). Reads were truncated at the 150-bp position, according to <the median quality score of <Q30 and the detected chimeric sequences were removed. Then, 8,121,517 filtered reads from 221 samples were considered for further analysis. After the sequence quality control step, the obtained amplicon sequence variants (ASVs) were assigned to taxonomy using a full-length pre-trained classifier SILVA_132 with OTUs clustered at 99%. Unassigned sequences, meaning ASVs with frequency <10 reads, were discarded, keeping 8,118,612 reads for the subsequent analysis. A rooted phylogenetic tree was constructed to measure phylogenetic diversity (Faith and UniFrac). ASVs were aligned with MAFFT (*Katoh & Standley, 2013*), and the resulting alignment was used to build a phylogenetic tree with FastTree (*Price, Dehal & Arkin, 2010*) software by using the align-to-tree-might-fast tree pipeline from the q2-phylogeny plugin.

### Diversity analysis

Library samples were rarefied to 2,900 reads to avoid unequal sample sizes and estimate alpha and beta diversity metrics. A rarefaction curve was performed sub-sampling on the processed data after deriving ASVs (post-ASV) to estimate species richness (alpha diversity) with the qiime diversity alpha-rarefaction plugin implemented in QIIME2 (Fig. S2) (*Bolyen et al., 2019*). Shannon, Chao1, and Faith's phylogenetic distance indexes estimated the samples' alpha diversity. Alpha diversity significance of Chao1 and Shannon indexes were performed with MicrobiomeAnalyst, a freely available online software (https://www.microbiomeanalyst.ca) (*Chong et al., 2020*; *Dhariwal et al., 2017*), using a Kruskal and Wilcoxon statistical test ($p < 0.05$) in the ASV set at the phylum level. Meanwhile, Faith's phylogenetic distance significance was performed in QIIME2 using the sub-sampled data with the plugin alpha-group-significance and the Kruskal–Wallis statistical test ($p < 0.05$) in the raw ASV at the feature level.

Beta diversity was calculated to estimate sample differences of pairs among tilapia gut microbial communities. Distance matrices were calculated using the Bray–Curtis dissimilarity, Weighted UniFrac distance, and Jensen–Shannon divergence. Bray-Curtis dissimilarity and Weighted UniFrac distance were performed with a sub-sampling of 2,900, using the plugin core-metrics-phylogenetic of QIIME2. Distance matrices were visualized using the principal coordinates analysis (PCoA) carried out by EMPeror from QIIME2. A pairwise comparison of the digestive tract beta diversity distance matrices was performed using the analysis of similarities (ANOSIM) within QIIME 2 to establish the degree of separation between the tested groups of samples. The statistical significance of the R statistic was assessed by 4,999 random permutations ($p < 0.05$) on the distance/dissimilarity matrix (*Clarke, 1993*). An R of 1 indicates complete separation, whereas an R of 0 indicates that the null hypothesis is true (*Chapman & Underwood, 1999*). A PCoA of the Jensen–Shannon divergence was also calculated at the phylum level with the statistical analysis ANOSIM, using the MicrobiomeAnalyst platform (https://www.microbiomeanalyst.ca) (*Chong et al., 2020*; *Dhariwal et al., 2017*). A PCoA of the Jensen–Shannon divergence was also calculated at the phylum level using the

MicrobiomeAnalyst platform (https://www.microbiomeanalyst.ca) (*Chong et al., 2020*; *Dhariwal et al., 2017*; *Lu et al., 2023*).

Abundance profiling of tilapia gut microbiota was performed as percentage abundance. Samples were merged into groups according to the sample type. The taxa resolution was set at the phylum level and small taxa with counts <20 were merged. In addition, Linear Discriminant Analysis (LDA) Effect Size (LEfSe) identified the key microbial taxa which are differentially abundant at the phylum level in Tilapia (*Oreochromis*) intestinal microbiota associated with the different additives included in their diet (*Segata et al., 2011*) and integrating the statistical significance with biological consistency (effect size) estimation. The LEfSe submodule within MicrobiomeAnalyst was used with the default settings of an FDR-adjusted *p*-value cut-off set to 0.05, and the log LDA cut-off at 2.0 (effect size) LEfSe analysis was performed with MicrobiomeAnalyst, a freely available online software (https://www.microbiomeanalyst.ca) (*Chong et al., 2020*; *Dhariwal et al., 2017*). Additionally, the prevalence of microorganisms at the phylum level across all the samples was estimated to define the core microbiome in the tilapia gut microbiota and performed with MicrobiomeAnalyst. The input table was performed using the relative abundances of each bioproject at the phylum level that comprises 90% of all the samples (Table S4).

## Correlation gut microbiota network analysis

Microbiome interaction networks were constructed *via* correlation values. To obtain the sparse correlation matrix for linear correlation among phyla in the tilapia gut microbiota among treatments (control, probiotic, prebiotic, and biofloc), we used the Pearson correlation coefficient after correcting for sample and taxon-specific biases with the Sparse Estimation of Correlations Among Microbiomes (SECOM) algorithm (*Lin, Eggesbo & Peddada, 2022a*). Biases considered with the SECOM model are the compositional, experimental, and zero excess bias (*Lin, Eggesbø & Peddada, 2022b*). Correlation networks were performed in the MicrobiomeAnalyst 2.0 platform (*Lu et al., 2023*).

## Functional prediction of the gut microbiome

The 16S rRNA amplicon sequencing data from bioprojects were processed to predict the functional potential of tilapia gut microbiota. Functional predictions were estimated using the Phylogenetic Investigation of Communities by Reconstruction of Unobserved States version 2 (PICRUSt2) (*Douglas et al., 2020*). PICRUSt2 aligned ASVs previously retrieved from QIIME2 to reference sequences using HMMER (*Finn, Clements & Eddy, 2011*); then, the resulting sequences were placed into a reference tree using EPA-NG and Gappa (*Barbera et al., 2019*). Also, predictions were normalized according to the bacterial 16S rRNA copies using castor from the hidden state prediction tool (*Louca & Doebeli, 2018*). The obtained prediction of metagenomic functional abundances was combined with descriptions from the Kyoto Encyclopedia of Genes and Genomes (KEGG) Orthology (KO) database at level 3. ASVs with an NSTI score >2 were removed from the final predictions. A heatmap was performed using the predicted functions of each bioproject using the KEEG level 3 table without descriptions. The input table was performed using the relative KO abundances of

each bioproject that comprise 90% of all the samples (Table S5). The heatmap was generated using a complete hierarchical clustering average linkage method with a one minus Pearson correlate matrix using the MORPHEUS web tool (Morpheus, Cambridge, MA, USA (https://software.broadinstitute.org/morpheus). In addition, a differential abundance (DA) analysis with the ALDEx2 method of the predicted functional profile was performed with the R package ggpicrust2 (*Yang et al., 2023*). The input table in the R package was the unstratified predicted metagenome of KO pathways generated by PICRUSt2.

## RESULTS

Alpha diversity indexes Chao1, Shannon, and Faith were unaffected by probiotics, prebiotics, or biofloc (Figs. 2 and 3), indicating that the gut microbiota of fish in terms of richness, evenness, and phylogeny remains relatively similar. Regarding beta diversity analyses performed by ANOSIM, no significant differences among the four groups were detected. In addition, PCoA estimated by Bray-Curtis (R = 0.019, $p = 0.33$), Unweighted UniFrac (R = 0.0042, $p = 0.38$), and Jensen–Shannon (R = 0.05 and $p = 0.792$) divergences did not show clear clustering or defined differentiation patterns between the studied groups (Figs. 4 and 5). For example, less than 23% and 43% of the variation was explained by axes 1, 2, and 3 in the Bray–Curtis and the Weighted Unifrac distances analyses, indicating that probiotics, prebiotics, or biofloc may not have a significant influence on the gut bacterial communities of fish either considering only the taxa abundance or the phylogenetic relatedness of such taxa. Also, principal coordinate analysis (PCoA) based on Jensen–Shannon divergence distance showed no clear differentiation pattern, with most of the samples (≥95%) located within the control area. Finally, no significant differences were detected when probiotics and prebiotics were separately compared with the control ($p > 0.05$).

Regarding taxonomic structure, similar profiles were observed with Proteobacteria, Fusobacteria, Actinobacteria, Firmicutes, Bacteroidetes, and Planctomycetes as the most representative phyla regardless of treatment (Fig. 6). However, effects on specific phyla were detected; for example, the LEfSe analysis ($p > 0.05$) revealed that Actinobacteria and Deinococcus-Thermus were influenced by prebiotic use, whereas the use of biofloc had a higher effect size on Proteobacteria, Bacteroidetes, Planctomycetes, Verrucomicrobia and Chlamydiae (Fig. 7). Fusobacteria and Chloroflexi showed an increase in the probiotic treatment. However, such individual changes do not significantly change the overall structure of the taxonomic profile.

A core microbiota could be defined across groups. At the phylum level, the tilapia core microbiota was dominated by Proteobacteria (31%), Fusobacteria (23%), Actinobacteria (19%), and Firmicutes (16%); however, other phyla were always present regardless of treatment, including Planctomycetes (1%), Chlamydiae (1%), Chloroflexi (1%), Cyanobacteria (1%), Spirochaetes (1%), Deinococcus-Thermus (1%), and Verrucomicrobia (1%), which served to construct a hypothetical polygon to visualize the variations in the taxonomic profile of tilapia (Fig. 8). At the genus level, *Cetobacterium* (23%), *Lactobacillus* (4%), *Legionella* (3%), *Lactococcus* (3%), *Rhodobacter* (2%), *Pelomonas*

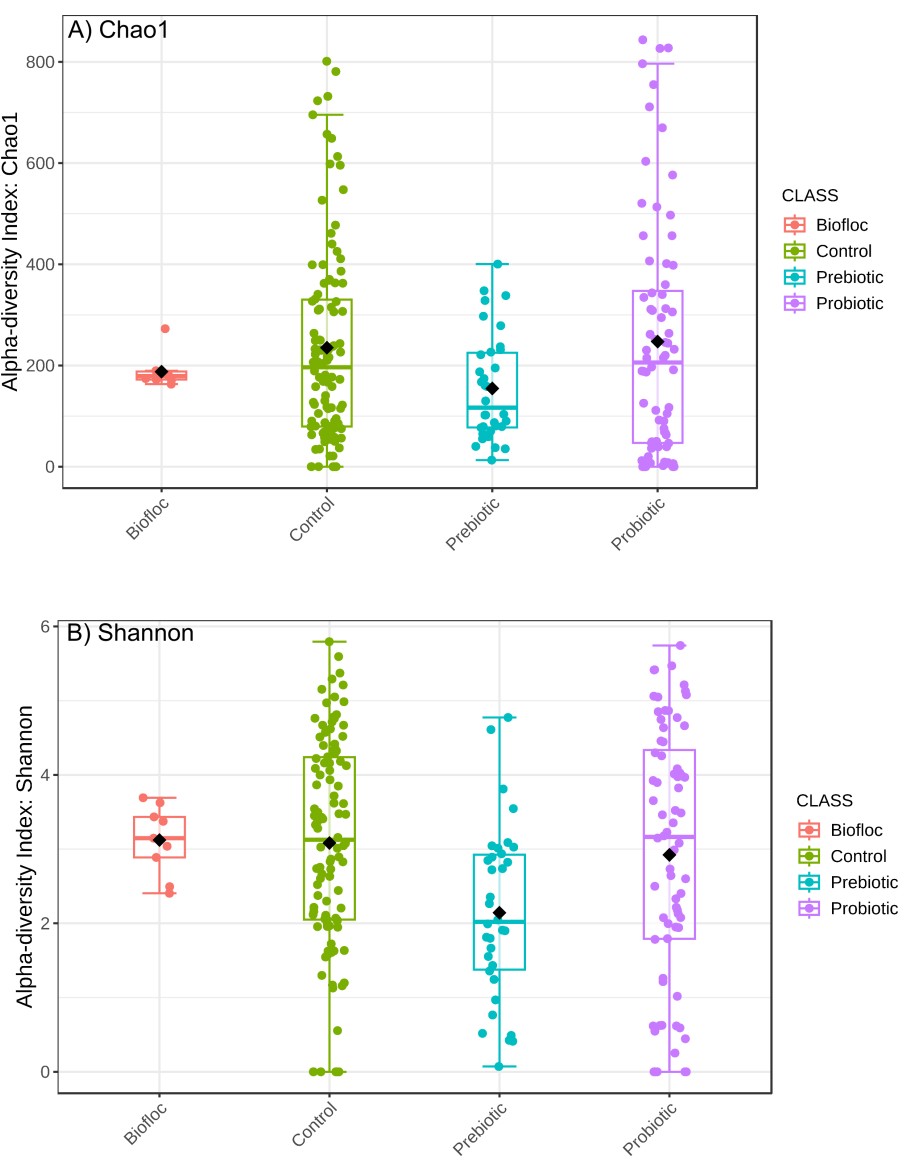

**Figure 2** **Alpha diversity of tilapia gut microbiota was estimated as Chao1 and Shannon indexes.** Alpha diversity analyses were estimated at the feature level as Chao1 and Shannon indexes to analyze the complexity of species diversity in the tilapia gut microbiota exposed to probiotic, prebiotic, and biofloc treatments. Fish not receiving any of the above treatments were grouped as control.

(2%), and *Streptococcus* (2%) were the most representative genera detected in all tilapia groups. Also, the core microbiome was defined by the phylum prevalence in all the samples. Proteobacteria was the most prevalent phylum among all the samples and also the phylum with the highest relative abundance. Other phyla remained stable among the samples; for instance, Firmicutes, Actinobacteria, and Bacteroidetes represented a 50% prevalence in the tilapia gut microbiota; such values are addressed in Table S6 (Fig. 9).

The Sparse Estimation of Correlations Among Microbiomes (SECOM) analysis was performed to assess the correlations between gut microbiota in tilapia. The significant

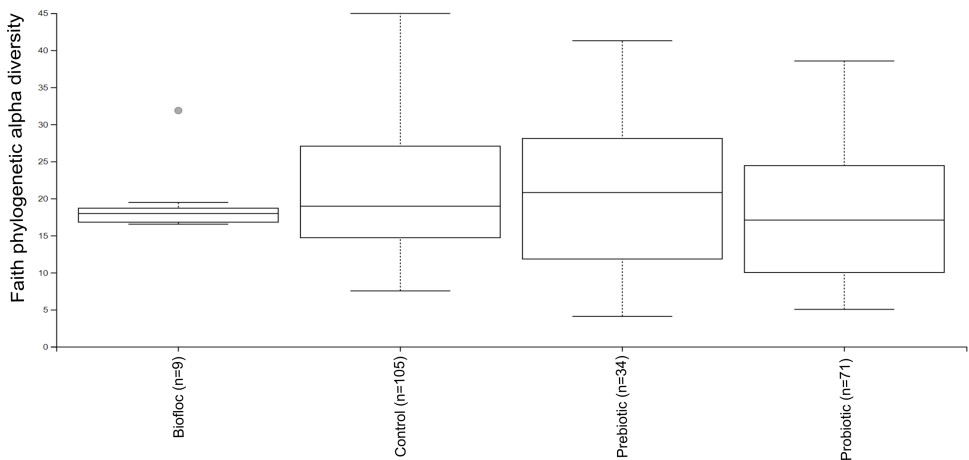

**Figure 3 Phylogenetic alpha diversity of tilapia gut microbiota.** Alpha diversity was estimated at feature-level with the faith phylogenetic diversity index of tilapia gut microbiota exposed to probiotic, prebiotic, and bioûoc treatments. Fish not receiving any of the above treatments were grouped as control.

correlations between bacterial phyla were presented in the correlation network (Fig. 10). Eight phyla were correlated among treatments, including, Actinobacteria, Bacteroidetes, Chloroflexi, Firmicutes, Fusobacteria, Planctomycetes, Proteobacteria, and Verrumicrobia, which showed a positive and negative correlation between each other. Interestingly, Chloroflexi was the phyla that showed the most correlations with seven phyla. Chloroflexi positively correlates with Bacteroidetes, Firmicutes, Fusobacteria, and Planctomycetes but negatively with Proteobacteria, Verrumicrobia, and Actinobacteria. Overall, a few positive correlations occurred among phyla; for instance, Chloroflexi and Planctomycetes registered the stronger positive correlation detected in tilapia gut microbiota with a value of 0.42; similarly, Proteobacteria and Verrumicrobia were the second most correlated phyla with a value of 0.38. At the same time, the highest negative correlation presented in the tilapia gut microbiota was between Proteobacteria and Chloroflexi, with a negative correlation value of −0.55, followed by Bacteroidetes and Actinobacteria with −0.47 (Table S7).

The heatmap of the predicted functional profiles from the tilapia gut microbiota inferred by PICRUSt2 does not present defined clusters among treatments (control, probiotic, prebiotic, and biofloc) (Fig. S3). Additionally, the results of the DA analysis of the functional predicted KEGG level 3 with the ALDEx2 method did not register significant features.

## DISCUSSION

The biological performance of the cultivated aquatic species can be favored using microbial consortia (biofloc), well-identified microbes (probiotics), or microbial-enhancing substances (prebiotics). Several reports have documented the influence of microbes and changes in environmental microbial composition on gut microbiota (*Abakari et al., 2021*; *Abdel-Ghany et al., 2020*; *Baumgartner, James & Ellison, 2022*). However, from a broader perspective, our results did not reveal significant differences in alpha and beta diversity,

## A) Bray-Curtis

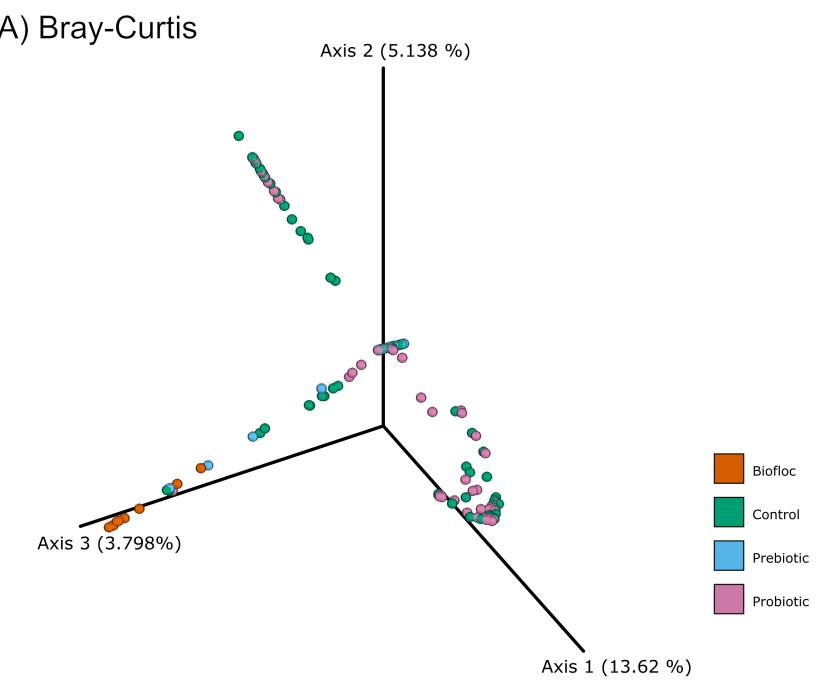

## B) Unweighted UniFrac

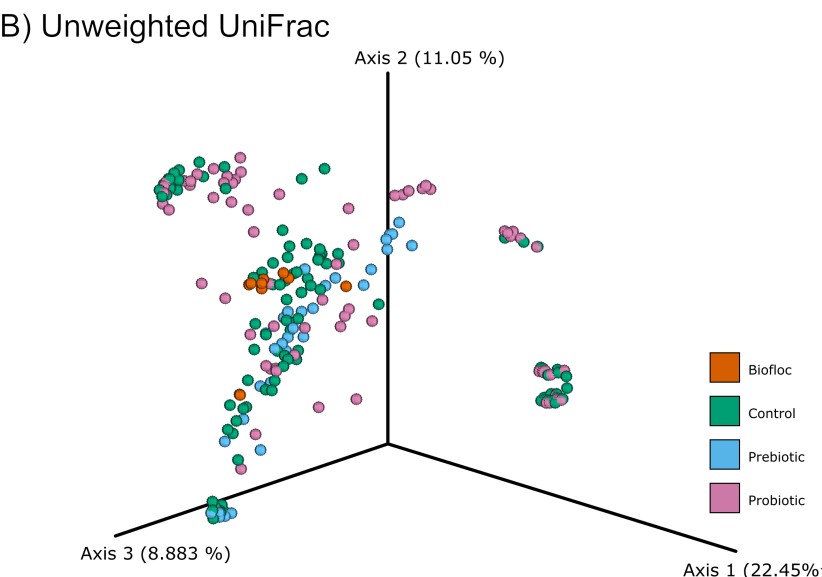

**Figure 4  PCoA of Beta diversity was calculated using the Bray–Curtis and Unweighted matrix distances.** Principal coordinate analysis (PCoA) was performed using the (A) Bray-Curtis (ANOSIM, $R = 0.019$, $p = 0.33$) and the (B) Unweighted Unifrac (ANOSIM, $R = 0.0042$, $p = 0.38$) distance matrix of the beta diversity of tilapia gut microbiota exposed to probiotic, prebiotic, and biofloc treatments. Fish not receiving any of the above treatments were grouped as control.

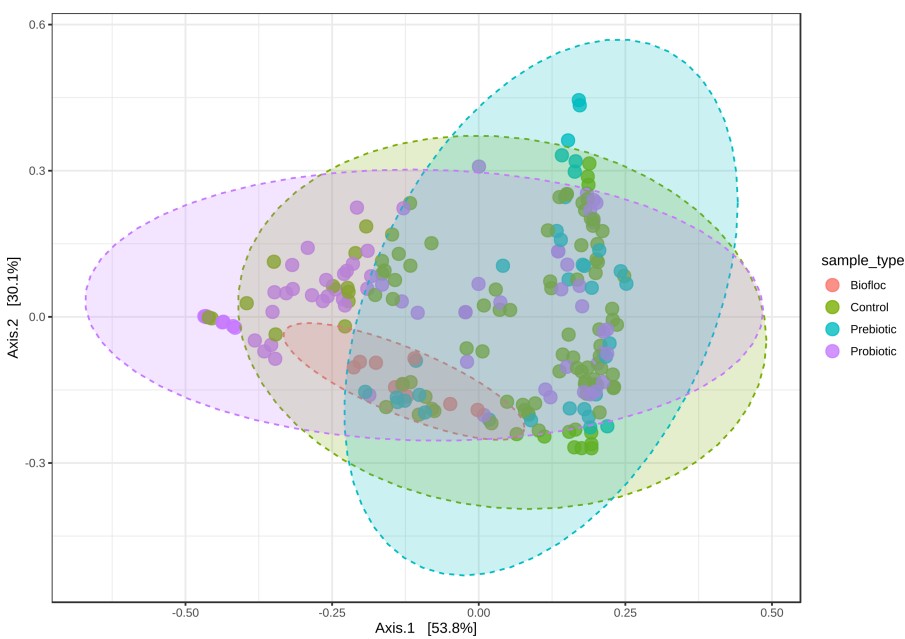

**Figure 5** **PCoA of Beta diversity was calculated using the Jensen–Shannon distance matrix.** Principal coordinate analysis (PCoA) based on the Jensen–Shannon divergence distance matrix shows the similarity of bacterial compositions of tilapia gut microbiota exposed to probiotic, prebiotic, and biofloc treatments. ANOSIM $R = 0.05$ and $p = 0.792$ . Fish not receiving any of the above treatments were grouped as control.

suggesting that modifications can only occur within a narrow range. It was impossible to define a pattern between the microbiota profiles of fish when they were or were not exposed to probiotics, prebiotics, and biofloc. However, the SECOM analysis showed networking within eight phyla in the tilapia gut microbiota, specifically, Actinobacteria, Bacteroidetes, Chloroflexi, Firmicutes, Fusobacteria, Planctomycetes, Proteobacteria, and Verrumicrobia as highly correlated phyla; however, this correlation is an expected outcome considering that all these constituted the core microbiota in tilapia. In addition, the functional predicted KEGG pathways of the 16S ARNr amplicon sequences from tilapia gut microbiota among treatments did not show significant changes. This steady state of the predicted functional profile remarks the functional redundancy importance in the tilapia gut ecosystem due to its implication in the community stability and resilience (*Biggs et al., 2020*). Overall, these results indicate that tilapia microbiota plasticity can withstand considerable microbiota variations of the intestinal tract to host different microbial taxa and their predicted functions. Despite PICRUSt2 provides accuracy and flexibility for marker gene metagenome inference, the predictions could be biased toward existing reference genomes (*Douglas et al., 2020*).

Although there are no studies in fish regarding the plasticity of the intestinal microbiota, this is recognized as a highly plastic entity in humans and animals, as it can be reconfigured in response to different environmental factors (*Candela et al., 2012*). This plasticity acts as a mutualistic configuration in which the microbiota can modify its functional and taxonomic profile caused by either intrinsic or extrinsic factors. In this case, it seems that

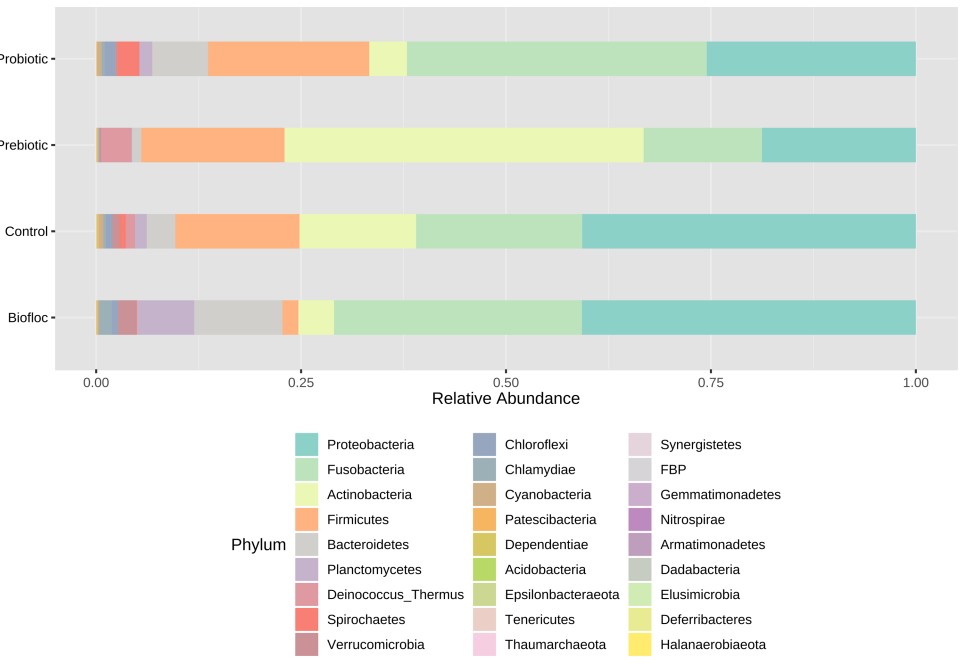

**Figure 6 Gut microbial composition at phylum level of tilapia.** Gut microbial composition at the phylum level of tilapia exposed to probiotic, prebiotic, and bioûoc treatments. Fish not receiving any of the above treatments were grouped as control.

using beneficial microbes or prebiotics does not modify the microbiota in a harmful way, as occurs in disease-associated fish microbiota profiles (*Medina-Félix et al., 2023*).

Even though the evidence shows that the microbiota responds plastically to beneficial microbes and prebiotics without leading to a substantive difference, some specific differences were detected. For instance, prebiotic use highly influenced Actinobacteria and Deinococcus-Thermus. Actinobacteria produce secondary metabolites acting against pathogenic microorganisms; the abundance of this phylum in fishes depends on the sediment composition and fauna residues in water (*Thejaswini et al., 2022*); in this case, prebiotics seem to favor Actinobacteria. Previous reports have documented Actinobacteria enrichments in the gut microbiota of other animals provided with similar prebiotics, including yeast cell walls high in beta-glucan and mannanoligosaccharides (*Vanden Abbeele et al., 2020*), galactooligosaccharides, xylooligosaccharides, and inulin (*Mitmesser & Combs, 2017*; *Wang et al., 2021b*). Regarding Deinococcus-Thermus, these bacteria are known for their resistance to extreme conditions (desiccation, high temperature, oxidation, radiation, oxidation). Whether the function of this phylum is still unclear in any gut microbiota, it is assumed (by genome sequencing) to participate in the metabolizing of sugars and probably in the elimination of organic and inorganic cell toxic components (*Méndez-Pérez et al., 2020*).

The linear discriminant analysis also revealed biofloc influencing Proteobacteria, Bacteroidetes, Planctomycetes, Verrucomicrobia and Chlamydiae, most of which are common in freshwater biofloc (*Liu et al., 2019*) and thus expected to influence the gut

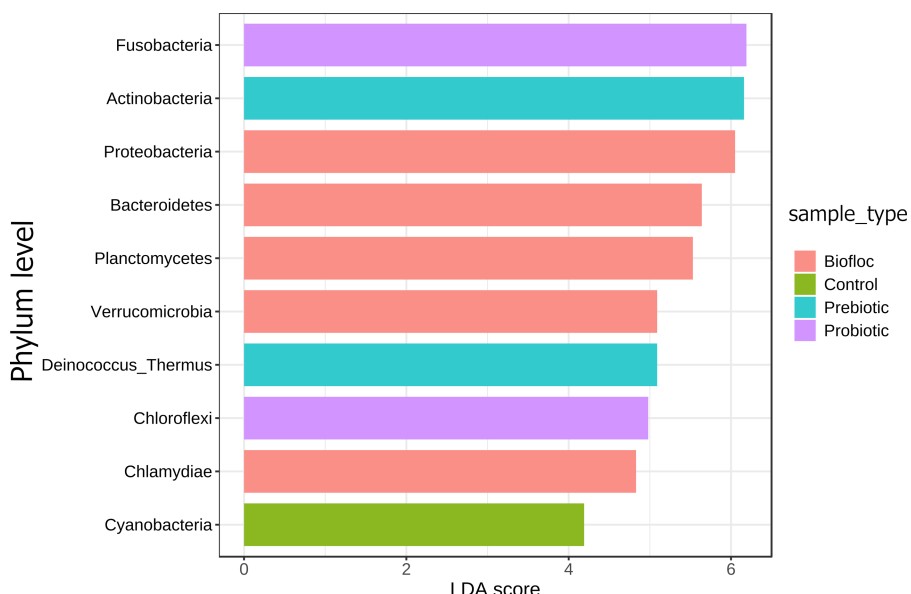

**Figure 7** **LEfSe analysis indicates diferentially abundant phyla.** Linear discriminant analysis effect size (LEfSe) analysis computed from phyla identified deferentially abundant (FDR-adjusted *p*-value cut-off set to 0.05) phyla in the analyzed gut microbiota of tilapia exposed and not exposed (control) to probiotics, prebiotics, and biofloc treatments. The top 10 enriched phyla in the gut tilapia microbiota are presented in the figure. Each different color represents the most abundant phyla by sample type.

microbiota; however, the concatenation of these changed did not influence the overall taxonomic profile of tilapia compared with the other studied groups. On the other hand, probiotics showed low or moderate effect size on most phyla. Although some of the bioprojects reported significant differences in the gut microbiota when additives were used, these changes were not different from the group concatenating all tilapia fish belonging to the respective controls suggesting that these changes occurred within an optimum interval delimited by the variations in the phyla forming the core of the gut microbiota.

Our results confirm previous evidence affirming that 80% of the gut microbiota of fish is formed by Proteobacteria, Fusobacteria, Actinobacteria, Firmicutes, and Bacteroidetes (*Yukgehnaish et al., 2020*). In this study, the concatenation of all analyzed projects revealed that these five phyla accounted for 93% of the relative abundance in the tilapia gut. Moreover, results revealed other phyla always detected in all groups, such as Planctomycetes, Deinococcus-Thermus, Spirochaetes, Chloroflexi, and Verrucomicrobia; therefore, these could be considered as minor members of the core microbiota of tilapia. In this regard, we propose the establishment of polygons formed and delimited by the interval of variance of the core microbiota in tilapia and other fishes, which may serve to determine if a variation in the gut microbiota is within or beyond safe limits and to compare gut microbiota profiles between taxa.

At more specific taxonomic levels, *Cetobacterium* (23%), *Lactobacillus* (4%), *Legionella* (3%), *Lactococcus* (3%), *Rhodobacter* (2%), *Pelomonas* (2%), and *Streptococcus* (2%) were the most representative genera, suggesting a relevant role at least in the balance of

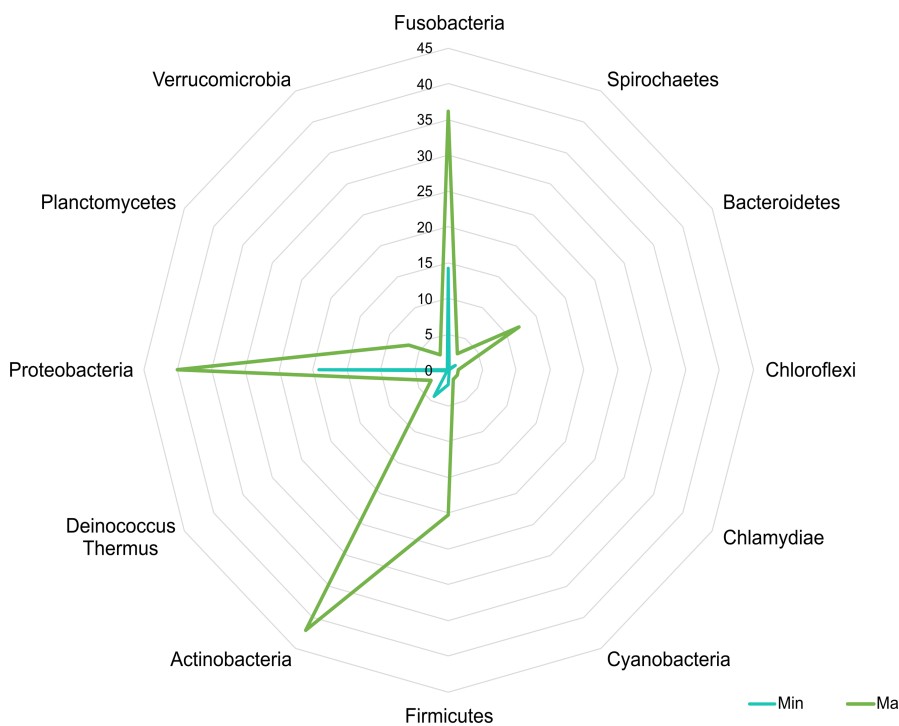

**Figure 8 Potential plasticity of tilapia gut microbiota.** The core microbial community that can plastically respond to feed additives exposure considering the minimum and maximum mean relative abundance of the phyla detected in the evaluated bioprojects. Numbers in the figure represent the percentage of relative abundance.

the gut microbiota, and providing information for therapeutic strategies for microbiota restoring purposes. Regarding the most abundant genera, *Cetobacterium*, this was also detected as the most abundant genera in carnivores like the hybrid striped bass, European bass, and red drum, in herbivores like the hybrid tilapia and flathead grey mullet, and omnivores like the common carp (*Ofek et al., 2021*). *Cetobacterium* is hypothesized to play beneficial roles in biochemical processes that contribute to glucose homeostasis and improve fish carbohydrate utilization (*Wang et al., 2021a*). *Lactobacillus* and *Lactococcus* are recognized as probiotics for fish (*Kuhlwein et al., 2014*; *Vargas-Albores et al., 2021*). *Legionella* has been identified as a pathogenic bacteria (*Olorocisimo et al., 2022*) but is frequently detected in fish. Although the biological role has not been elucidated (*Bereded et al., 2022*) it is probably a pathobiont contributing with significant functions to the microbiota but acting as a pathogen under specific circumstances. *Rhodobacter* species are considered potential antibiotic substitutes in crustacean and fish aquaculture; for instance, protein supplementation obtained from *Rhodobacter* inhibits the propagation of intestinal opportunistic pathogens, while improving growth, immune response, antioxidant capability, and survival in shrimp (*Liao et al., 2022a*; *Liao et al., 2022b*).

In the end, despite some of the individual projects reported microbiota modifications when using additives, the conglomeration of information from multiple projects suggests that although additives may influence the microbiota, these modifications remain within

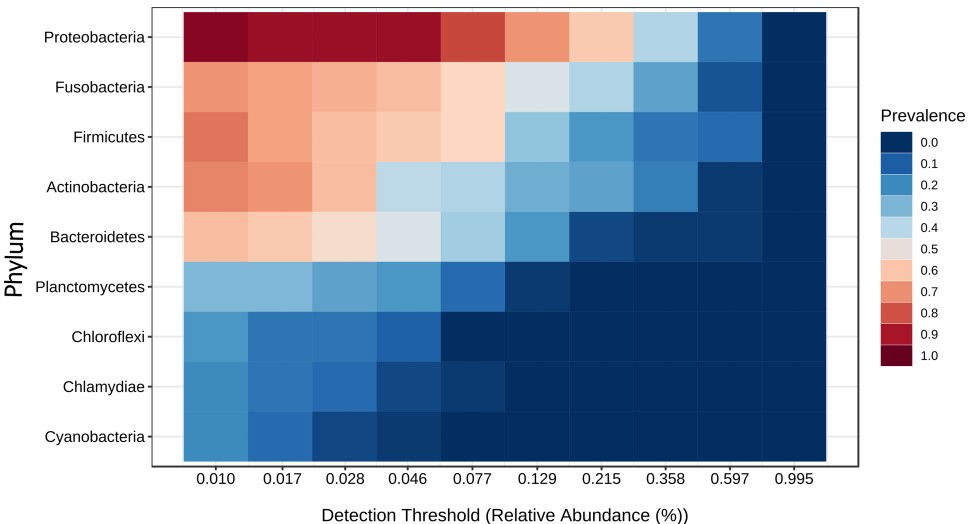

**Figure 9 Core microbiome base on the phylum prevalence.** Heatmap of the tilapia gut microbiota core at phylum level which include the most prevalent taxa among all the treatments (probiotics, prebiotics and bioflocs). The *x*-axis represents the relative abundance detection from lower to higher abundance values. Color shading indicates the prevalence of each bacterial family among samples for each abundance threshold. As we increase the detection threshold, the prevalence decreases.

an optimal range of variation delimited by the plasticity of the intestinal microbiota. Finally, it is possible that this same pattern could occur with other factors that impact the microbiota.

## CONCLUSIONS

This meta-analysis suggests little variations in the structure and composition of gut microbial communities among tilapia gut microbiota exposed to feed additives (probiotics, prebiotics, and biofloc) from the integrated 221 samples from different tilapia gut microbiota studies. Despite technical and host factor biases can influence the obtained results, as expected in meta-analytic approaches, some patterns were defined and contributed to establishing the composition and variations of the tilapia gut microbiota while defining a host-adapted core microbiota, which included the phyla Proteobacteria, Fusobacteria, Actinobacteria, Firmicutes, and Bacteroidetes. In this regard, we also conclude that the gut microbiota of tilapia is a plastic component that can vary as a response to probiotics, prebiotics, and biofloc addition. At the same time, tilapia gut microbiota is an adaptive and probably resilient component with a wide dynamic range that seems to allow a considerable optimal range of variation; therefore, modifications in the taxonomic profile caused using feed additives may be safe for tilapia.

Additionally, the results provide perspectives for developing therapeutic manipulations using the signature microorganism of the tilapia gut microbiota. Consequently, tilapia with great dysbiosis could modify or regenerate their microbiota configuration. Moreover, it is necessary to assess the gut microbiota adaptability strategies and relations among the microorganisms to comprehend the complex gut ecosystem.

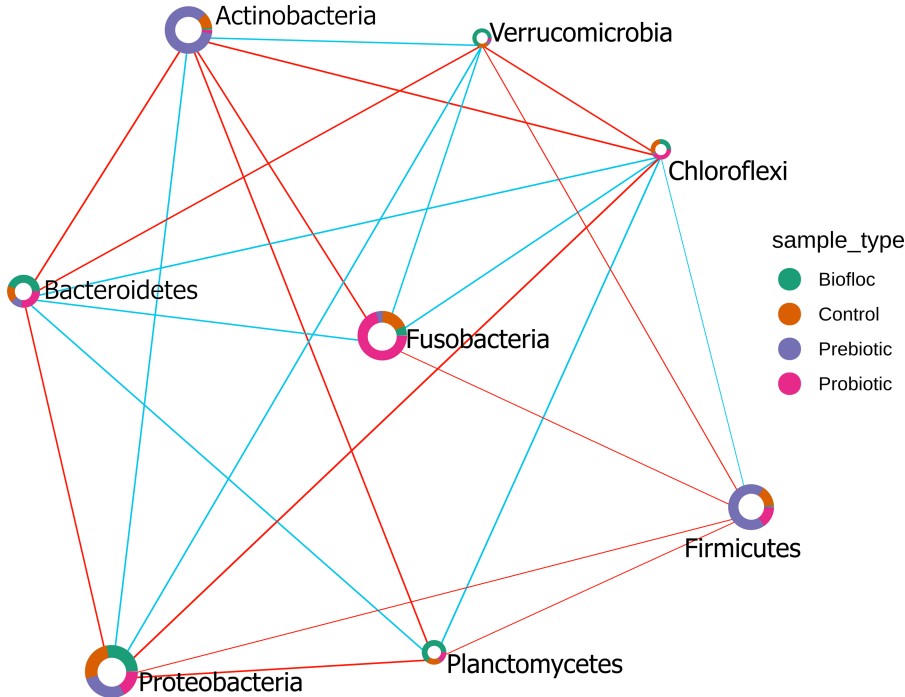

**Figure 10** SECOM correlation network analysis applied to tilapia gut microbiome at the phylum level. SECOM correlation network analysis applied to tilapia gut microbiome at the phylum level. Estimation of Correlations Among Microbiomes (SECOM) network analysis at the phylum level of the tilapia gut microbiota reveals significant interactions. Each node represents a phylum level, and its size is based on the number of connections to the phylum. Different colors in the node indicate the phylum proportion by sample type (control, probiotic, prebiotic, and biofloc). The edge thickness is equivalent to the correlation values. Blue edges represent positive correlations and red edges represent negative correlations.

# ACKNOWLEDGEMENTS

The authors thank Azucena Santacruz and Sheyla Acosta for their contribution to the manuscript format and figures.

### Funding

The authors received no funding for this work.

### Competing Interests

The authors declare there are no competing interests.

### Author Contributions

- Marcel Martinez-Porchas conceived and designed the experiments, analyzed the data, authored or reviewed drafts of the article, and approved the final draft.
- Aranza Preciado-Álvarez performed the experiments, prepared figures and/or tables, and approved the final draft.
- Francisco Vargas-Albores performed the experiments, analyzed the data, authored or reviewed drafts of the article, and approved the final draft.
- Martina Hilda Gracia-Valenzuela analyzed the data, authored or reviewed drafts of the article, and approved the final draft.
- Francesco Cicala performed the experiments, authored or reviewed drafts of the article, and approved the final draft.
- Luis Rafael Martinez-Cordova conceived and designed the experiments, authored or reviewed drafts of the article, and approved the final draft.
- Diana Medina-Félix performed the experiments, prepared figures and/or tables, and approved the final draft.
- Estefania Garibay-Valdez analyzed the data, prepared figures and/or tables, and approved the final draft.

## Data Availability

The accession numbers of the included bioprojects are available in the Supplementary Files.

## Supplemental Information

Supplemental information for this article can be found online at http://dx.doi.org/10.7717/peerj.16213#supplemental-information.

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
