# Peer review of "Microbiota plasticity in tilapia gut revealed by meta-analysis evaluating the effect of probiotics, prebiotics, and biofloc"

_PeerJ, doi:10.7717/peerj.16213_

## Round 0.1 · original submission · Major Revisions

Please respond to the reviewers' comments.

**Language Note:** The review process has identified that the English language must be improved. PeerJ can provide language editing services - please contact us at copyediting@peerj.com for pricing (be sure to provide your manuscript number and title). Alternatively, you should make your own arrangements to improve the language quality and provide details in your response letter. – PeerJ Staff

Reviewer 1 ·

Basic reporting

The manuscript titled 'Tilapia gut microbiota plasticity revealed by meta-analysis evaluating the effect of probiotics, prebiotics, and biofloc' has addressed most of the questions by previous reviewers very well. However, minor revision is still needed prior to its acceptance by the journal.

Experimental design

comments for Introductions:
69-70 “Water quality management is another challenge because tilapia require high-quality water.” Citations are needed to support this statement.

comments for Methods:
170-171: “After removing low-quality scores, taxonomy was assigned to amplicon sequence
variants (ASVs) using a pre-trained classifier SILVA_132 with OTUs clustered at 99%.” Please provide more specific information about the criteria for removing low-quality scores, and clarify what the pre-trained classifier SILVA_132 entails.

213-216: “The LEfSe submodule within MicrobiomeAnalyst was used with the default settings of an FDR-adjusted p-value cut-off set to 0.1 and the log LDA cut-off at 2.0 (effect size) LEfSe analysis was performed with MicrobiomeAnalyst” FDR-adjusted p-value cut-off should be set to 0.05.

SECOM method is not well explained in the manuscript.

Validity of the findings

no comment

Additional comments

Comments for Figures:
Figure 4 and Figure 5: P values for statistics in PCoA should be put in Figures or the legends.
Figure 7: Are these phyla enriched in Biofloc or prebiotic? Please provide the information in the figure legend. FDR should be set to 0.05 to search the differential abundant phyla.
Figure 8: What do the numbers represent in the figure? Please provide the information in the legend.
Figure 10: The network figure is very confusing and the colors in the circles are not defined in the figure. SECOM method is not well explained in the manuscript.

·

Basic reporting

Dear author, I have reviewed the paper "Are mining companies ready to assess and manage biodiversity-related risks?", which aimed to evaluate the effect of probiotics and other additives ingested by tilapia on their gut microbiota through a meta-analysis of several bioprojects studying the gut microbiota of tilapia exposed to food additives (probiotic, prebiotic, biofloc).

 The structure of the paper is generally very good. The authors present a clear summary where an introduction, the problem, methodology, results and conclusions are briefly addressed. Regarding the introduction, it puts in context from the presentation of the subject of study, to the problem and the objective of analysis of the study. The methodology and review processes used are clearly described. The results could be presented in a better way by sectioning the two key points of the study, which are narrated in the last paragraph of the introduction.

Experimental design

It would be interesting if the authors could better explain the relationship in the 60 articles with the projects that were considered, perhaps for the readers this is not clear. Including or improving a methodological diagram would be the solution. Where the process is captured from the search, analysis and correlation.

Validity of the findings

The results are presented clearly and precisely. The discussion and conclusions address the objectives of the study. If anything, I would say that the authors should generate two opening paragraphs in the conclusions that address each of the two central points of the study. Then if the last paragraph on limitations and future studies from their results is perfect.

Reviewer 3 ·

Basic reporting

the manuscript in general written using sufficient literature references and the idea of the study is clear
But, English editing is needed

Experimental design

The experiment was designed well
and meet the Ain and scope of the journal
Methods described sufficiently

Validity of the findings

Authors illustrated the results clearly using the suitable method of statistical analysis
and the results were supported throughout the discussion in a good manner.
Conclusion was well stated and showed the aim of designing the study .

Additional comments

Authors should write the key words after the abstract
Rewrite the title of manuscript

English editing is needed
Line 382: should be “It is “
Revise the references according to the style of journal

Reviewer 4 ·

Basic reporting

I found the manuscript to be well-written, with informative nature and valuable insights into tilapias gut microbiota.

Experimental design

I have some uncertainties regarding the utilization of multiple hypervariable regions in the analysis of tilapia gut microbiota and how these regions were aligned and further analyzed all together. I would have preferred the authors to had focused on a specific region of the 16S gene, and to had exclude for the current analysis the rest data. Concentrating on a selected hypervariable region can offer more focused insights and enhance the comparability with existing research data.

Validity of the findings

After receiving feedback from previous reviewers, the authors incorporated a PICRUSt analysis. However, the presentation of the data lacks informativeness. The authors should include descriptions of the most elevated pathways identified in the gut microbiota and their possible contribution. This would provide valuable insights into the functional potential of the microbial community.

Additional comments

It appears that the majority of my comments and concerns have been addressed in the point-by-point response letter to the previous reviewers. However, my main concern remains focused on the utilization of multiple 16S regions analyzed together. I'm uncertain if this concern can be adequately addressed solely through further justification and clarifications from the authors. It might require more detailed explanations and potentially a reevaluation of the approach to ensure the validity and reliability of the results.

---

## Round 0.2 · Minor Revisions

Please respond to the minor comments of the reviewer.

Reviewer 1 ·

Basic reporting

The authors have addressed most of the concerns raised by the reviewers. However, there is still one area that needs to be addressed before it can be accepted for publication.

Experimental design

It’s still unclear how the authors conducted the metabolic analysis and functional prediction, given that they appear to have only used 16s rRNA sequence data in this study. If the authors used metagenomic samples in the study, they should point that out. If the authors solely relied on 16s rRNA sequencing data for the functional prediction, they should acknowledge the limitation in the discussion and exercise caution in their wording regarding functional prediction.

Validity of the findings

no comment

---

## Round 0.3 · accepted · Accept

Your paper is accepted in PeerJ
Congratulation